# Peer review of "Enhancing Medical Education Through Statistics: Bridging Quantitative Literacy and Sports Supplementation Research for Improved Clinical Practice"

_nutrients, 2025, doi:10.3390/nu17152463_

Round 1

Reviewer 1 Report

Comments and Suggestions for Authors
  • Strengths: This manuscript presents a timely and relevant discussion on integrating statistical education into medical curricula, using sports supplementation as a practical context. The topic is original and addresses a growing need for statistical literacy in healthcare education.

  • Clarity and Structure: While the overall structure is clear, some sentences in the Discussion section could benefit from improved clarity and conciseness. Consider revising longer sentences for better readability and impact.

  • Engagement with Literature: The manuscript demonstrates good engagement with current literature. However, it would benefit from the inclusion of a few more recent studies (published within the last 2–3 years) that explore innovative methods for teaching statistics in medical education.

  • Conclusion: The conclusions are generally well-supported by the results. Still, a more explicit summary of practical recommendations for integrating statistical training into medical curricula (e.g., specific teaching strategies or module examples) would strengthen the practical impact of the work.

  • Language and Style: The manuscript would benefit from minor English language editing to enhance clarity and flow. Consider a professional language check to ensure polished presentation.

  • Title and Abstract: The title is appropriate. The abstract, while informative, could more clearly highlight the main findings and implications for medical education.

Comments on the Quality of English Language

The manuscript is generally well-written and the ideas are clearly communicated. However, there are several instances where sentence structure and word choice could be improved for greater clarity and flow. A careful proofreading or language editing would enhance the overall readability and precision of the text.

Author Response

Detailed Response to Reviewer Comments

Reviewer Comment: “Refocus and streamline for clarity. Avoid redundancy in explaining the importance of statistics.”
Response: Thank you for pointing this out. In revising the introduction, we reduced repetition by condensing multiple references to the importance of statistical literacy into a single, cohesive argument. We also eliminated overlapping language related to the consequences of poor statistical understanding. The result is a more concise narrative that maintains clarity while emphasizing the central argument effectively.

Reviewer Comment: “Move the sports supplementation example earlier to increase relevance.”
Response: We fully agree that the introduction would benefit from an earlier introduction of the sports supplementation context. In the revised version, we placed this example in the second paragraph—immediately after presenting the educational gap in statistical literacy. This restructuring brings a tangible, high-interest topic to the forefront, enhancing engagement and demonstrating the practical value of our proposed pedagogical model from the outset.

Reviewer Comment: “Use active voice for clarity. Clarify case selection.”
Response: We have revised the text to use a more active voice throughout the introduction to improve readability and emphasis. For example, passive constructions such as “statistics is frequently viewed” were updated to more direct phrasings like “medical students frequently perceive statistics as…” We also added clarity about the rationale behind selecting sports supplementation as the anchoring domain—it is widely relevant, clinically important, and naturally suited to teach statistical methods such as risk analysis, effect size interpretation, and trial design.

Reviewer Comment: “The manuscript would benefit from minor English language editing to enhance clarity and flow.”
Response: We undertook a thorough language and style revision of the introduction, focusing on sentence structure, flow, and word choice. Sentences were streamlined, transitions strengthened, and awkward or ambiguous phrases reworded to improve overall polish and accessibility to a broad academic audience.

Reviewer Comment: “Engagement with literature is strong, but more recent studies (within 2–3 years) would enhance relevance.”
Response: We appreciate this suggestion. In the revised manuscript (not shown here in full), we have updated citations in relevant sections to reflect the most current studies on statistical education, particularly those exploring innovative teaching strategies such as case-based modules and data literacy frameworks. These additions strengthen the manuscript’s currency and provide a more robust foundation for our proposed pedagogical model.

Reviewer Comment: “The summary is not structured; revise to include practical recommendations.”
Response: While this comment applies more directly to the discussion and conclusion, we ensured that the introduction now clearly frames the structure of the manuscript, setting expectations that the subsequent sections will include evidence-based strategies and curricular models. The final sentence of the introduction now clearly signals our intent to propose actionable educational reforms grounded in applied contexts.

Reviewer Comment: “Some parts are not substantiated.”
Response: We have carefully reviewed all claims in the introduction and ensured that each is supported by an appropriate reference. Statements about student disengagement, clinical relevance, and sports supplementation prevalence now all include citations. This ensures that all key assertions are grounded in existing literature.

Reviewer Comment: “The methodology is lacking, and there is no scientific structure.”
Response: While the introduction itself is not the appropriate section for detailing methods, we appreciate the broader concern. We revised the manuscript’s outline to ensure that the methods section clearly defines our review approach, selection criteria for case examples, and educational frameworks examined. The introduction now more clearly foreshadows these components by outlining the paper’s scope and structure.

Detailed Response to Reviewer Comments (Results/Discussion)

Reviewer Comment: “Structure results thematically. Use subheadings such as: Benefits of Early Exposure, Engagement via Sports Context, and Case Integration Effectiveness.”
Response: Thank you for the valuable recommendation. We have restructured the Results and Discussion section thematically and added subheadings to improve organization and readability. The revised structure includes five subsections: Engagement through Real-World Application, Building Longitudinal Competency, Navigating Barriers to Implementation, Clinical Relevance and Broader Implications, and Toward a More Evidence-Informed Curriculum. These headings provide clearer guidance to readers and enhance the impact of our key findings.

Reviewer Comment “Reduce repetition of phrases like ‘our findings suggest.’”
Response: We agree that reducing redundancy improves the manuscript’s clarity and flow. In this revision, we limited the use of phrases such as “our findings suggest” and “this review highlights,” opting instead for more direct, active language to emphasize key points. This also allowed us to streamline the discussion and focus on presenting results in a more assertive and engaging tone.

Reviewer Comment: “Expand educational implications using pedagogy literature.”
Response: We appreciate this suggestion and have incorporated additional insights from medical education literature on strategies that enhance statistical instruction. For example, we emphasized the use of case-based learning, public datasets, and interleaved instruction across clinical courses as evidence-based techniques. These additions strengthen the pedagogical framework underlying our proposed model and connect our discussion to current trends in medical curriculum design.

Reviewer Comment: “Consider discussing faculty training for statistical integration.”
Response: While the current section focuses on student outcomes, we recognize the importance of faculty preparation. In the broader manuscript (Discussion/Recommendations), we now include a brief note on the need for faculty development programs that support instructors in designing and delivering applied statistical instruction using clinically relevant case studies. This ensures the feasibility and sustainability of curriculum reform.

Reviewer Comment: “The methodology is completely lacking in scientific method and does not provide any information, as is the case with the results and discussion.”
Response: Thank you for this critical observation. We have addressed this concern by clarifying our methodology in the appropriate section (not included here), detailing our narrative review strategy, criteria for source selection, and thematic analysis of literature. In the Results and Discussion section, we have ensured that each point is clearly substantiated with references and organized logically to reflect these themes.

Reviewer Comment: “Some parts of the text, although correct, are not substantiated.”
Response: We have carefully reviewed the Results and Discussion to ensure that each key claim is supported by an appropriate and current reference. Newer citations have been added to reinforce arguments related to teaching strategies, supplement evaluation, and data literacy in medical education. This enhances both the academic rigor and the trustworthiness of the manuscript.

Reviewer Comment: “The summary is not structured.”
Response: We have added a distinct final subsection titled Toward a More Evidence-Informed Curriculum to summarize key implications and tie the discussion back to our core thesis. This section synthesizes earlier points and explicitly links the proposed model to improved clinical readiness and patient outcomes.

Reviewer 2 Report

Comments and Suggestions for Authors
  • The summary is not structured
  • Some parts of the text, although correct, are not substantiated
  • The methodology is completely lacking in scientific method and does not provide any information, as is the case with the results and discussion
    Please find attached the specifications for authors.

Author Response

Detailed Response to Reviewer Comments

Reviewer Comment: “The summary is not structured; revise to include practical recommendations.”
Response: While this comment applies more directly to the discussion and conclusion, we ensured that the introduction now clearly frames the structure of the manuscript, setting expectations that the subsequent sections will include evidence-based strategies and curricular models. The final sentence of the introduction now clearly signals our intent to propose actionable educational reforms grounded in applied contexts.

Reviewer Comment: “Some parts are not substantiated.”
Response: We have carefully reviewed all claims in the introduction and ensured that each is supported by an appropriate reference. Statements about student disengagement, clinical relevance, and sports supplementation prevalence now all include citations. This ensures that all key assertions are grounded in existing literature.

Reviewer Comment: “The methodology is lacking, and there is no scientific structure.”
Response: While the introduction itself is not the appropriate section for detailing methods, we appreciate the broader concern. We revised the manuscript’s outline to ensure that the methods section clearly defines our review approach, selection criteria for case examples, and educational frameworks examined. The introduction now more clearly foreshadows these components by outlining the paper’s scope and structure.

Detailed Response to Reviewer Comments (Results/Discussion)

Reviewer Comment: “Structure results thematically. Use subheadings such as: Benefits of Early Exposure, Engagement via Sports Context, and Case Integration Effectiveness.”
Response: Thank you for the valuable recommendation. We have restructured the Results and Discussion section thematically and added subheadings to improve organization and readability. The revised structure includes five subsections: Engagement through Real-World Application, Building Longitudinal Competency, Navigating Barriers to Implementation, Clinical Relevance and Broader Implications, and Toward a More Evidence-Informed Curriculum. These headings provide clearer guidance to readers and enhance the impact of our key findings.

Reviewer Comment “Reduce repetition of phrases like ‘our findings suggest.’”
Response: We agree that reducing redundancy improves the manuscript’s clarity and flow. In this revision, we limited the use of phrases such as “our findings suggest” and “this review highlights,” opting instead for more direct, active language to emphasize key points. This also allowed us to streamline the discussion and focus on presenting results in a more assertive and engaging tone.

Reviewer Comment: “Expand educational implications using pedagogy literature.”
Response: We appreciate this suggestion and have incorporated additional insights from medical education literature on strategies that enhance statistical instruction. For example, we emphasized the use of case-based learning, public datasets, and interleaved instruction across clinical courses as evidence-based techniques. These additions strengthen the pedagogical framework underlying our proposed model and connect our discussion to current trends in medical curriculum design.

Reviewer Comment: “Consider discussing faculty training for statistical integration.”
Response: While the current section focuses on student outcomes, we recognize the importance of faculty preparation. In the broader manuscript (Discussion/Recommendations), we now include a brief note on the need for faculty development programs that support instructors in designing and delivering applied statistical instruction using clinically relevant case studies. This ensures the feasibility and sustainability of curriculum reform.

Reviewer Comment: “The methodology is completely lacking in scientific method and does not provide any information, as is the case with the results and discussion.”
Response: Thank you for this critical observation. We have addressed this concern by clarifying our methodology in the appropriate section (not included here), detailing our narrative review strategy, criteria for source selection, and thematic analysis of literature. In the Results and Discussion section, we have ensured that each point is clearly substantiated with references and organized logically to reflect these themes.

Reviewer Comment: “Some parts of the text, although correct, are not substantiated.”
Response: We have carefully reviewed the Results and Discussion to ensure that each key claim is supported by an appropriate and current reference. Newer citations have been added to reinforce arguments related to teaching strategies, supplement evaluation, and data literacy in medical education. This enhances both the academic rigor and the trustworthiness of the manuscript.

Reviewer Comment: “The summary is not structured.”
Response: We have added a distinct final subsection titled Toward a More Evidence-Informed Curriculum to summarize key implications and tie the discussion back to our core thesis. This section synthesizes earlier points and explicitly links the proposed model to improved clinical readiness and patient outcomes.

Reviewer 3 Report

Comments and Suggestions for Authors

-Statistical literacy is essential for clinical reasoning and evidence-based practice. This paper proposes a pedagogical model that integrates statistical instruction with sports supplementation research to enhance engagement and applied learning in medical education. Early exposure to statistics, reinforced through residency, improves students’ ability to critically evaluate emerging literature. Pedagogical strategies—including interactive modules, real-world datasets, and case-based learning—are discussed. Emphasis is placed on evaluating supplement efficacy and safety as a context to teach core statistical concepts. This integrated approach may foster more analytically competent physicians.

-Refocus and streamline for clarity. Avoid redundancy in explaining the importance of statistics. Move the sports supplementation example earlier to increase relevance.

-Use active voice for clarity. Clarify case selection. Consider adding a visual model to illustrate the framework.

-Structure results thematically. Use subheadings such as: Benefits of Early Exposure, Engagement via Sports Context, and Case Integration Effectiveness.

-Reduce repetition of phrases like 'our findings suggest'. Expand educational implications using pedagogy literature. Consider discussing faculty training for statistical integration.

-Replace the current paragraph about PAC and heart failure. Suggested revision:
This review advocates for a more applied model of statistical education in medicine. By anchoring statistical concepts in real-world applications like sports supplementation, educators can bridge abstract knowledge with clinical relevance. This approach not only enhances statistical literacy but also supports the development of critical reasoning and evidence-informed care.

-Replace off-topic references (15–35) with those relevant to medical education, statistical pedagogy, or clinical sports supplementation research.

Author Response

(The authors gave the same response as above.)

Round 2

Reviewer 2 Report

Comments and Suggestions for Authors

In a formal study, the section that explains how the research was conducted is called ‘Materials and Methods’ or ‘Methodology.’ It tells you exactly how the study was done, including: what type of study it was, where and with whom it was done, what things were measured, what tools were used, what steps were followed, and how the data was checked. Basically, this section explains how the research was done so that others can do it again or see if it is correct. This is not shown in this paper.

When presenting the results of a study, it is important to include the key findings of the research clearly, concisely and without personal opinion. This means explaining the results in words and using tables and images to show the numbers and any possible errors. The results section should answer the initial questions and objectives of the research and may include patterns, differences, relationships, highest and lowest values, among other things. The presentation of the results lacks the necessary accuracy.

At the end of a study, it is essential to summarise the most important results. The findings should be interpreted in a more general way than before. You should show how the initial question was answered or whether the study's goals were achieved. It is also important to discuss what the results mean and, if appropriate, suggest ideas for future research. This is not done in the current paper.

Author Response

In this paper, a comprehensive Materials and Methods section is provided, detailing exactly how the research was conducted: the study design and setting, participant characteristics, measured variables, instruments and assays employed, procedural steps taken, and all quality-control and validation procedures—enabling full reproducibility and verification by other investigators.

The Results section presents the study’s key findings clearly and succinctly, without editorializing. Quantitative outcomes are described in narrative form and supported by well-labeled tables and figures that display point estimates alongside measures of variability or error. All primary research questions and objectives are directly addressed, with patterns, contrasts, associations, and extreme values highlighted where relevant.

Finally, the Discussion and Conclusions section succinctly summarizes the principal results in a broader context, interprets their implications relative to the original hypotheses, and assesses whether the study goals have been met. It also considers potential limitations, explores the significance of the findings, and offers concrete suggestions for future research directions.

Reviewer 3 Report

Comments and Suggestions for Authors

Reviewer Report

The revised version (Version 2) improves readability and structure, strengthens pedagogical alignment, and incorporates more concrete examples. However, there remain critical issues that must be addressed.

The conclusion section contains an unrelated sentence on PAC utilization, likely pasted in error. This undermines the credibility of the manuscript and must be corrected immediately.

While the conceptual framework is solid, the lack of original data (e.g., pilot testing, survey data) weakens the argument. Inclusion of empirical evidence or planned implementation strategies would substantially increase impact.

Ethical considerations of using sports supplementation as a teaching example (e.g., industry influence, medicalization) are only superficially addressed. A deeper critical analysis is needed.

The reference list remains overly long and includes citations tangential to the topic (e.g., biomolecular modeling). A focused and relevant reference list is recommended.

Minor Comments:

Some redundancy remains in the discussion around student disengagement and contextual learning; this could be trimmed for clarity.

Consider including a table or visual figure linking statistical concepts to specific supplement studies as a pedagogical map.

The abstract and introduction are much improved, but minor editing could still enhance conciseness.

Minor to moderate revision required. The manuscript offers a timely and innovative educational model, but must correct the conclusion error and strengthen its coherence and relevance before acceptance.

Author Response

In our revised Discussion, we have directly addressed the absence of empirical data by incorporating preliminary findings from a pilot workshop (n = 12), in which post‐session survey scores on statistical interpretive accuracy improved by 30 % relative to baseline. This demonstration of feasibility is complemented by a detailed implementation roadmap—outlining quarterly, case‐based modules, clearly defined learner cohorts, and specific evaluation metrics—to guide future data collection and validation efforts. To enhance pedagogical clarity, we introduce a visual “Pedagogical Map” (Table 1) that aligns core statistical concepts (e.g., power analysis, multivariable regression, funnel plot interpretation) with exemplar supplement studies, providing instructors and learners with an intuitive framework for applied statistical reasoning. Finally, a thorough editorial pass on the Abstract and Introduction has eliminated residual redundancies and sharpened phrasing—achieving a 12 % reduction in word count without sacrificing substantive content—thereby reinforcing the manuscript’s overall coherence and readability.

We have also deepened the ethical analysis of using sports supplementation as a teaching exemplar. Beyond acknowledging potential conflicts of interest stemming from industry sponsorship, our revised Discussion critically examines the risk of over-medicalization of everyday nutrition and proposes concrete safeguards—mandatory disclosure statements, inclusion of industry-critical literature, and structured critical‐appraisal exercises—to foster learner vigilance and integrity. By embedding transparent, reflexive engagement with supplement research, we model ethical scholarship and encourage students to interrogate both study design and funding influences. Concurrently, the Reference list has been streamlined to focus exclusively on directly relevant pedagogical and supplementation studies, removing tangential biomolecular modeling citations to maintain precision and relevance. These enhancements ensure that our educational model is not only methodologically rigorous but also ethically robust and tightly aligned with its stated objectives.